# Sea star wasting syndrome reaches the high Antarctic: Two recent outbreaks in McMurdo Sound

**Amy L. Moran**[1]*, **Rowan H. McLachlan**[2], **Andrew R. Thurber**[2,3]

**1** School of Life Sciences, University of Hawai'i at Mānoa, Mānoa, Hawaii, United States of America, **2** Department of Microbiology, College of Science, Oregon State University, Corvallis, Oregon, United States of America, **3** College of Earth, Ocean, and Atmospheric Sciences, Oregon State University, Corvallis, Oregon, United States of America

* morana@hawaii.edu

**Data Availability Statement:** All relevant data are within the paper and its Supporting Information files.

**Funding:** Funded by NSF-OPP-1745130 to ALM and NSF-OPP-2046800 to ART. United States

## Abstract

Sea star wasting syndrome (SSWS) can cause widespread mortality in starfish populations as well as long-lasting changes to benthic community structure and dynamics. SSWS symptoms have been documented in numerous species and locations around the world, but to date there is only one record of SSWS from the Antarctic and this outbreak was associated with volcanically-driven high temperature anomalies. Here we report outbreaks of SSWS-like symptoms that affected ~30% of individuals of *Odontaster validus* at two different sites in McMurdo Sound, Antarctica in 2019 and 2022. Unlike many SSWS events in other parts of the world, these outbreaks were not associated with anomalously warm temperatures. Instead, we suggest they may have been triggered by high nutrient input events on a local scale. Although the exact cause of these outbreaks is not known, these findings are of great concern because of the keystone role of *O. validus* and the slow recovery rate of Antarctic benthic ecosystems to environmental stressors.

## Introduction

Marine diseases are of increasing concern as ocean conditions alter with global climate change [1–6]. One of the diseases that has had considerable recent impacts is starfish wasting disease or sea star wasting syndrome (SSWS). This disease occurs in sudden and sometimes widespread outbreaks that can cause high levels of mortality in affected populations [7–12]. Outbreaks of SSWS-like symptoms in wild sea star populations have been noted for over a hundred years and on most continents [8]. The largest known event in the past decade began in 2013, when mass occurrences of SSWS were recorded on both the east and west coasts of North America. More than 20 species of sea star were affected, and the event caused the near-complete loss of a top predator species in much of its geographic range [5,11].

Because sea stars are important predators in benthic ecosystems, outbreaks of SSWS can drive major and potentially long-lasting changes in community structure in affected regions [5,9,10,13–15].

National Science Foundation Office of Polar Programs, https://www.nsf.gov/div/index.jsp?div=OPP. The funders had no role in study design, data collection and analysis, decision to publish, or preparation of the manuscript.

**Competing interests:** The authors have declared that no competing interests exist.

SSWS is characterized by the development of lesions on the aboral surface of individuals, often accompanied by deflation of the body, twisting of arms, and progressive disintegration of the integument. While not always fatal, the progression of the disease can be very rapid, with as little as 7–10 days between symptom onset and death [11,16], and it can affect large percentages of the population at particular sites [11,12,17]. In both lab and field, SSWS has been linked to changes in sea star habitat including elevated organic material, drops in oxygen availability, and increases in temperature, among others [7]. Anomalously warm seawater temperatures, in particular, have been implicated in SSWS outbreaks in the field [5,8,11,17–20] and regional effects of SSWS on asteroid populations in the 2013–2104 outbreak on the west coast of North America were stronger and more long-lasting in low-latitude, warmer areas compared to colder areas [5]. Recent studies suggest that the proximate cause may be oxygen depletion in the diffusive boundary layer at the surface of the animal, created through microbial activity which is enhanced by warmer temperatures and increased organic inputs [7,21].

The Southern Ocean surrounding Antarctica is one of the coldest and most stable water masses on Earth, and contains a diverse and abundant assemblage of sea stars [22]. Despite the widespread occurrence of SSWS elsewhere on the globe [8], only one previous occurrence has been reported in the Southern Ocean. This outbreak was observed in 2013–2014 in the flooded caldera of Deception Volcano, located close to the northern end of Antarctic Peninsula, by Núñez-Pons et al. [19]. The disease was only observed in one species, the valvatid sea star *Odontaster validus* (Koehler, 1906), and it affected up to 10% of individuals at some sites within the caldera. The timing of the outbreak coincided with a period of geothermal activity that caused numerous episodic temperature bursts up to 10˚C above normal winter ocean temperatures [19,23]. Other than this one event, to our knowledge, SSWS has never been observed in the Southern Ocean–perhaps due to the relative inaccessibility of the environment, which has led to an under-sampling of sea star diversity in the region generally [22].

Recently, during routine surveys at two locations in McMurdo Sound, Antarctica, we observed two separate outbreaks of SSWS that affected localized populations of the same species, *O. validus*. The first outbreak was recorded in September of 2019 at the McMurdo Intake Jetty, a long-established research diving site just offshore of the United States Antarctic Program's McMurdo Station. The second outbreak was observed at Cinder Cones, another established nearshore research diving site approximately 5 km (shoreline distance) from the Intake Jetty, in October of 2022. *Odontaster validus* is a widespread and common sea star that is a keystone predator and detritivore in Antarctic benthic ecosystems [24–28], thus repeated observations of SSWS in this species were concerning. We performed surveys on SCUBA to assess the frequency and severity of the disease in the outbreak areas and in areas immediately adjacent. Dive teams also looked for evidence of SSWS at multiple other sites around McMurdo Sound in 2019, 2021, and 2022. Lastly, in 2020, we re-surveyed the 2019 outbreak site at the McMurdo Jetty site to assess population-level effects as well as the duration of the outbreak.

## Methods

Quantitative surveys of SSWS incidence were performed at the two sites where we observed outbreaks of SSWS-like symptoms; the McMurdo Intake Jetty and Cinder Cones. Both are established research dive sites that have been visited regularly by science divers for more than 30 years. Several other dive sites with populations of *O. validus* were also visited during the same period and observationally evaluated for presence/absence of SSWS. All dive sites are shown in Fig 1 and Table 1. No organisms were taken during this project. No permits were required for site access because the study sites were not within any Antarctic Specially Protected Areas (ASPA) or Antarctic Specially Managed Areas (ASMA) as defined by Annex V to

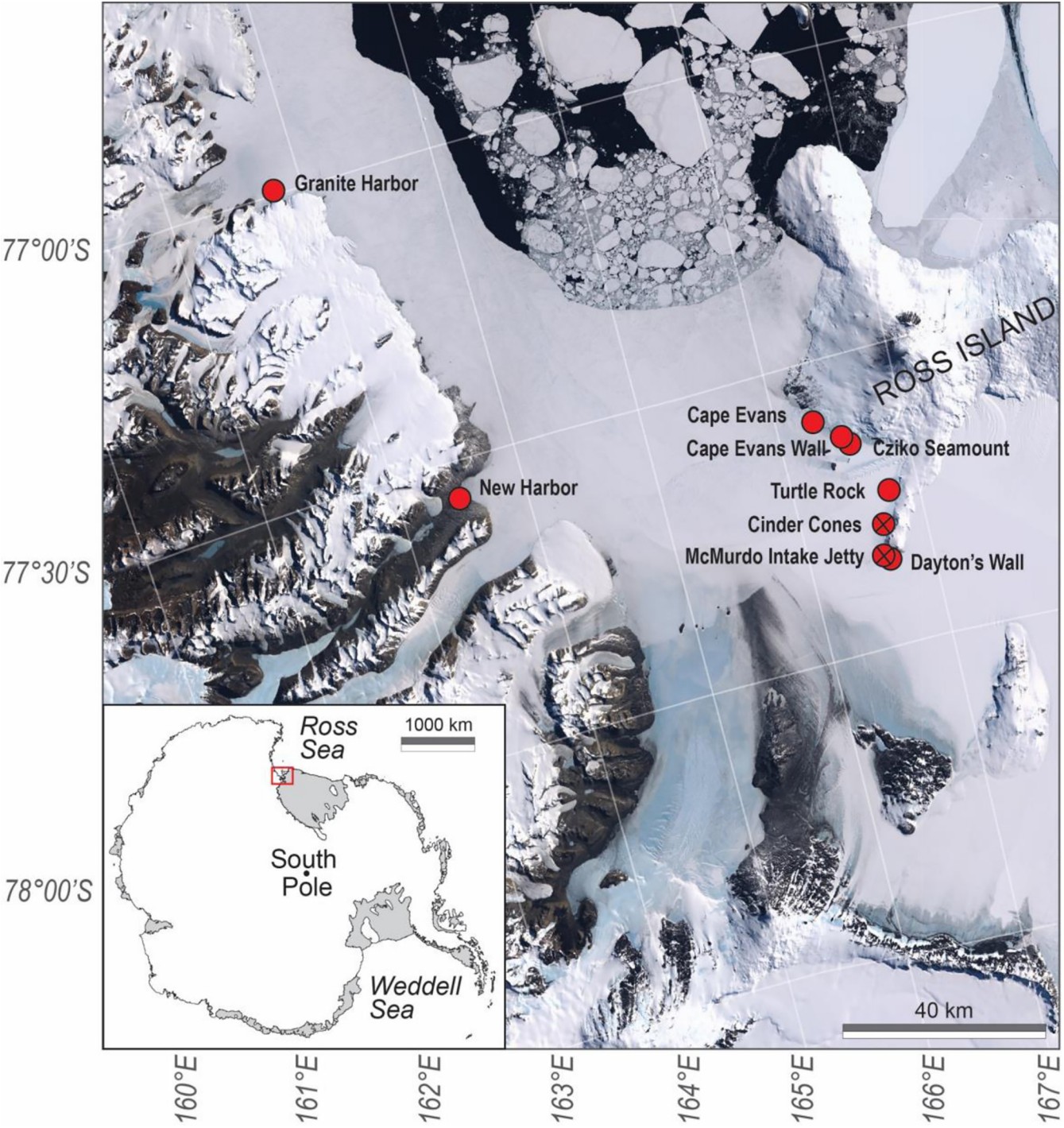

**Fig 1. Map of dive sites.** Map showing the McMurdo Sound region, Ross Sea, Antarctica, and the location of all dive sites visited in this study (red circles). All sites had populations of *Odontaster validus*. Open red circles indicate no diseased animals were observed at a particular site; red circles with an x indicate the locations of the two sites where SSWS outbreaks were observed and documented. The map used in Fig 1 is modified from the Landsat Image Mosaic of Antarctica (LIMA) Project and is in the public domain.

**Table 1. Dive site metadata.** Localities, years, survey types and presence/absence of symptoms of sea star wasting syndrome (SSWS) for nine research dive sites that were monitored for outbreaks of SSWS.

| Dive site name | GPS coordinates | Years visited | Type of survey | SSWS present |
|---|---|---|---|---|
| McMurdo Intake Jetty | 77˚ 51.069' S, 166˚ 39.855' E | 2019, 2020, 2021, 2022 | Transects, observational | 2019, 2020 |
| Cinder Cones | 77˚ 47.998' S, 166˚ 40.241' E | 2016, 2022 | Video transect (archive), transects | 2022 |
| Cape Evans | 77˚ 38.064' S, 166˚ 24.876' E | 2019, 2021, 2022 | Observational | None seen |
| Cape Evans Wall | 77˚ 38.407' S, 166˚ 31.068' E | 2019, 2021, 2022 | Observational | None seen |
| Dellbridge Seamount | 77˚ 38.927' S, 166˚ 31.873' E | 2019, 2021, 2022 | Observational | None seen |
| Dayton's Wall | 77˚ 51.1949' S, 166˚ 39.7830' E | 2019, 2021, 2022 | Observational | None seen |
| Turtle Rock | 77˚ 44.639' S, 166˚ 46.175' E | 2019, 2021, 2022 | Observational | None seen |
| New Harbor | 77˚ 34.268' S, 163˚ 30.685' E | 2019, 2021, 2022 | Observational | None seen |
| Granite Harbor | 77˚ 00.715' S, 162˚ 51.515' E | 2019, 2021, 2022 | Observational | None seen |

the Environment Protocol, Agreed Measures for the Conservation of Antarctic Fauna and Flora, Antarctic Treaty Consultative Meeting.

## McMurdo Intake Jetty

Surveys at the Intake Jetty were conducted by divers in October and November of 2019 and in October and November of 2020 on two 90 x 2 meter transects that ran parallel to rock wall of the Jetty in the immediate area of the observed outbreak. One transect followed the south side of the Intake Jetty ("JS") and the second ran along the adjacent north side ("JN"). During surveys, each diver scored every *Odontaster* in their survey area for presence of SSWS symptoms and the degree of disease progression, on a scale of 0 (no visible disease) to 4 (nearly complete dissolution of soft body parts) (Figs 2 and 3, Table 2). To avoid double-counting, three separate transects were made at different depths; one on the rocks of the Jetty (~20 m depth), one on the rock-mud interface (~21 m depth), and one on the mud (~22 m depth). JN and JS were each surveyed twice along the same transects in 2019 at intervals of approximately one month; JS on October 17 and November 10, and JN on October 21 and November 21. The same transects were surveyed again in 2020 on November 28 (JS) and November 30 (JN). No formal surveys were conducted at JN or JS after 2020, but research divers visited the Intake Jetty multiple times in 2021 and 2022 and kept an eye out for diseased animals.

A chi-square test of independence was performed to compare the frequencies of asymptomatic animals (stage 0) to symptomatic animals (pooled stages 1–4) between 2019 and 2020. The same transect lines were surveyed twice in 2019, so to avoid potential double-counting of animals, for 2019 we only used data from the first sampling date on each side of the Jetty (10/17/19 for JS, 10/19/19 for JN). For both 2019 and 2020, data from JN and JS were pooled within year. All data are shown in S1 Table. The analysis was performed in JMP Pro 15.2.0 (SAS Institute Inc., 2019).

## Cinder Cones

In 2022, because the outbreak appeared to be mostly localized to the region of a methane seep that ran parallel to the shore at ~11 m depth (habitat described in Thurber et al. 2020), three

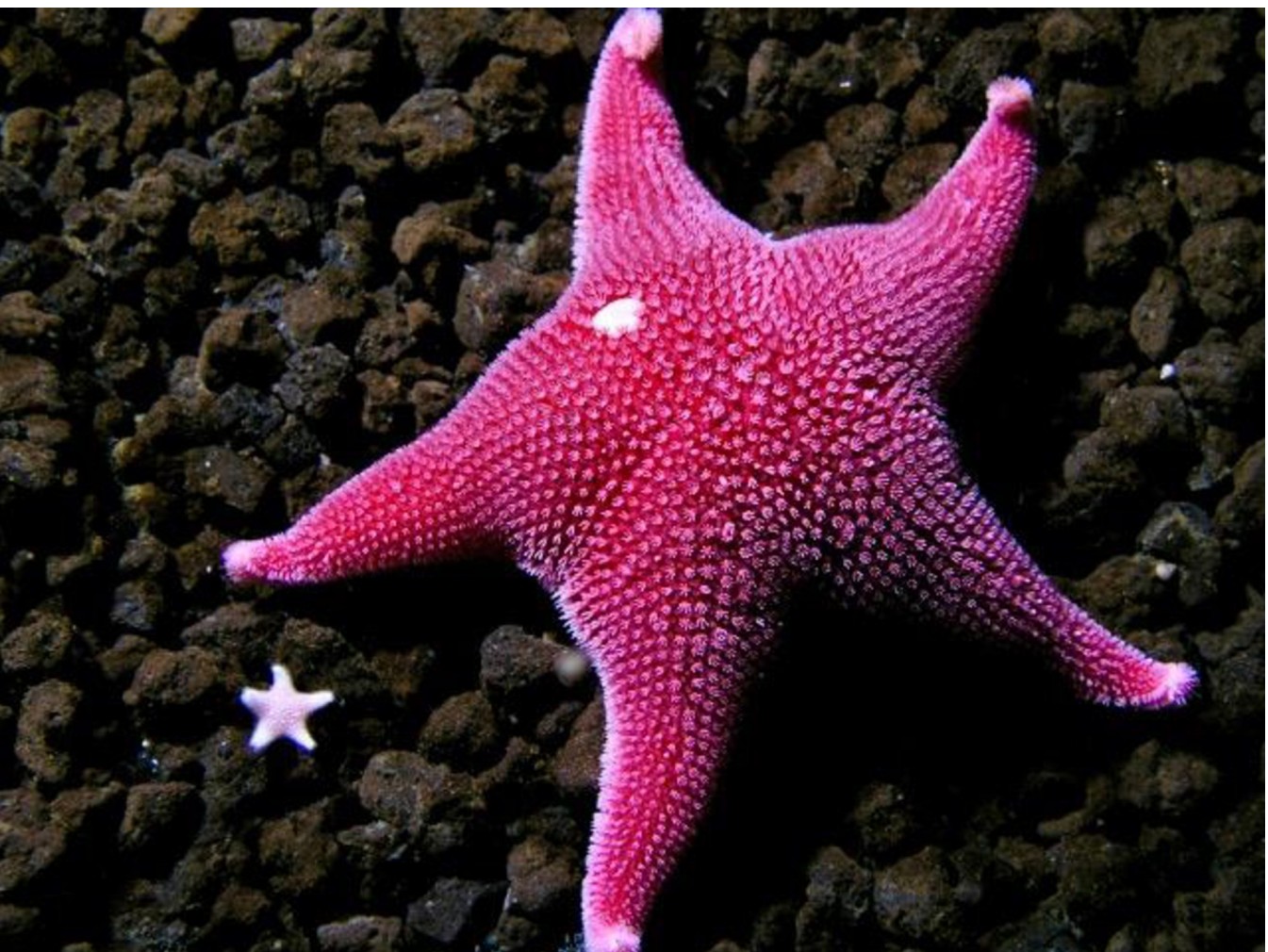

**Fig 2. Healthy individual of *Odontaster validus* (stage 0, no disease).** McMurdo Intake Jetty, Antarctica, 2019. Photograph by Rob Robbins, used with permission under CC BY 4.0.

parallel transects were established: 1) inside the seep area at 10 m depth where there were numerous small anemones in the genus *Edwardsia* (hereafter referred to as the "*Edwardsia* field"), 2) above the *Edwardsia* field at ~8 m depth, and 3) below the *Edwardsia* field at ~13 m. Each transect was 50 x 2 meters. All *O. validus* within the transect areas were counted and categorized for disease symptoms and progression by one of the divers (Robbins) who performed the Intake Jetty surveys in 2019 and 2020, using the same criteria.

Cinder Cones was also visited for other purposes in 2016 and video transects were performed along the seep at the same location as the 2022 transects. This archival footage from 2016 was reexamined by the authors in 2022 to look for past evidence of SSWS on the methane seep.

## Other sites

Throughout the austral summers of 2019–2022, dives for other purposes were conducted at six sites in addition to the Intake Jetty and Cinder Cones. Four of these were on the west side of Ross Island, one was on the continental (west) side of McMurdo Sound, and one was a

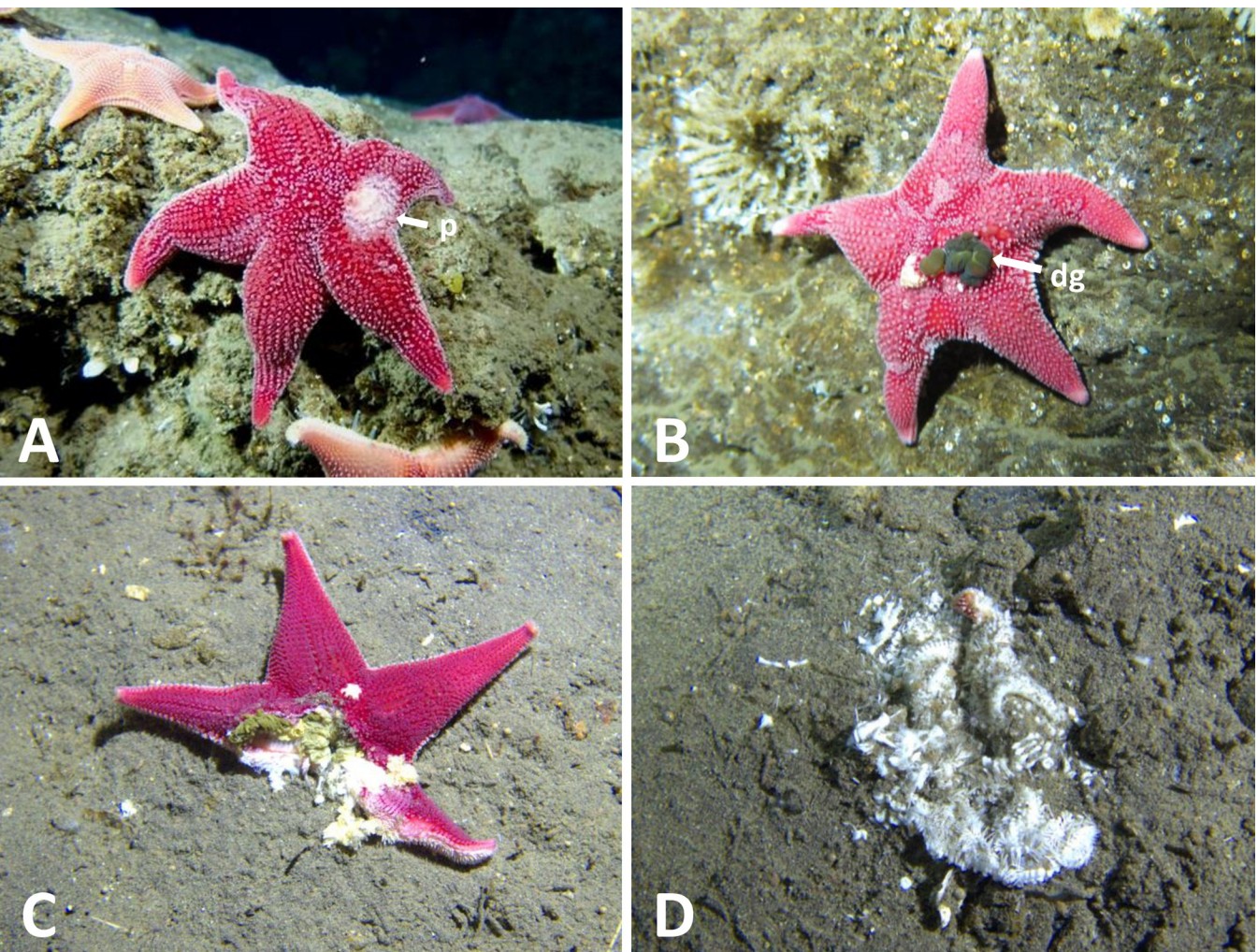

**Fig 3. Symptomatic individuals of *Odontaster validus*.** Stages described in Table 1. A, Stage 1; B, stage 2; C, stage 3; D, stage 4. p = light patch, dg = digestive gland protruding through a hole in the body wall. McMurdo Intake Jetty, Antarctica, 2019. Photographs by Rob Robbins, used with permission under CC BY 4.0.

continental site further to the north in the Ross Sea (Fig 1, Table 1). No formal surveys were conducted at these sites, but divers were always on the lookout for symptomatic animals. Most of these sites have been visited in multiple previous years without specifically looking for SSWS.

**Table 2. Categories of disease severity for surveys at the McMurdo Intake Jetty and Cinder Cones.** Animals were scored 0–4 based on the presence/absence and severity of symptoms.

| Stage | Criteria |
|---|---|
| 0 | No visible discoloration or damage to the body surface |
| 1 | Small patches of pale discoloration on aboral surface, no full perforations of body wall |
| 2 | Discoloration and small perforations of the body wall, with digestive gland protruding |
| 3 | Substantial damage, with large holes in the body wall and/or partial loss of arms or oral disc |
| 4 | Near-complete disintegration of soft tissue |

## Results

All raw survey count data, percentages, and densities are provided in the (S1 Table).

### McMurdo Intake Jetty

In 2019, 35.4 ± 8.4 and 27.1 ± 9.4% (mean ± SE of the surveys at the three different depths) of *O. validus* showed visible signs of SSWS on the two survey dates (October and November, respectively). The highest disease rate in any transect at JS was 49.5% and the lowest was 14.0%. JN had a lower percentage of affected animals (10.7 ± 6.2 and 8.4 ± 2.4%), with a high of 22.9% and a low of 2.4% (in 2019). The majority of affected animals at JS were in stages 2–4 (80.7 and 56.8% for the two dates), reflecting more severe disease, while at JN most affected animals were at stage 1 (76.5 and 88.2%). Overall, densities of sea stars were ~2x higher at JS compared to JN.

In 2020, densities of *Odontaster* had dropped by approximately 50% at JS and by > 80% at JN. Disease rates were considerably lower than in 2019, with only two affected animals out of 186 at JS and none (out of 34) at JN. A chi-square test showed that there was a significant relationship between disease frequency and year, $\chi^2$ (1, N = 909) = 64.24, p <0.0001. Symptomatic animals were significantly less frequent in 2020 compared to 2019.

No diseased animals were seen in 2021 or 2022 at either JS or JN.

### Cinder Cones

Disease rates in the *Edwardsia* field were 29.4 and 29.1% for the two different surveys in 2022, with 92 and 95% of affected animals in categories 2–4. Disease rates were considerably lower in the single surveys conducted above the field (8 m) and below it (13 m), at 4.8 and 8%, respectively. The proportions of animals in categories 2–4 were 71.4% (8 m depth) and 76.7% (13 m depth).

### Other sites

Observations by research divers showed no SSWS outbreaks at any other site between 2019 and 2022. Occasionally, divers would report a single animal that appeared to be damaged, but in no case were multiple animals affected nor did the symptoms clearly resemble SSWS rather than mechanical damage from a different source (e.g., predation, ice scour).

## Discussion

Outbreaks of SSWS-like symptoms in sea star populations have been reported for > 100 years and at multiple sites in both the Northern and Southern hemispheres (reviewed in [8]) yet to date, there is only one previous record from the Southern Ocean [19], from the northerly end of the Antarctic Peninsula. The outbreaks we report here from two sites in McMurdo Sound are, therefore, the most southerly documented occurrences of SSWS that we know of. We cannot say these are the first occurrence of SSWS in the high Antarctic, because unlike the west coast of North America, the Antarctic nearshore environment is mostly inaccessible and unobserved. However, the McMurdo Intake Jetty is the most extensively and frequently-dived site in the Antarctic. According to the 34-year United States Antarctic Program McMurdo Station dive record (1989-present), the McMurdo Intake Jetty was visited every year between 1989 and 2023, with over 3,500 logged dives at the site. Cinder Cones was visited 21 out of those 34 years, for a total of 797 dives. Due to the frequency and amount of research activity at both sites, particularly the McMurdo Intake Jetty, it seems unlikely that previous outbreaks of

SSWS at this site would have been missed. *Odontaster validus* has also been frequently studied and collected in McMurdo Sound over several decades (e.g., [24–26,28–37]).

Similar to the event at Deception Island, symptoms of SSWS in McMurdo Sound were localized to particular dive sites and were not noted in any other species of asteroid besides *O. validus*. *O. validus* was by far the most common species at our sites, though, and other species were not surveyed in a systematic way. Therefore, SSWS in other species cannot be ruled out. However, the McMurdo Sound outbreaks were considerably more severe, in that almost 50% of animals were affected in some surveys (compared to 10% in [19]) and a high proportion of the animals were in later, and clearly terminal, stages of the disease. The disease symptoms we observed in McMurdo Sound were similar to the major SSWS event in the NE Pacific, with total soft-part disintegration of many animals. Densities of *Odontaster* were considerably lower at the Intake Jetty in 2020 compared to 2019, but we cannot confidently attribute this to SSWS because the decline in numbers was greater on the north side (JN), which had a far lower proportion of diseased animals in 2019. This could mean that the JN experienced high mortality from SSWS in the winter of 2020 when no divers were in the water to observe; alternatively, *O. validus* are mobile and could have migrated out of the immediate area of the surveys. The outbreak at the Intake Jetty was largely confined to 2019 with very few symptomatic animals observed in 2020 and none in subsequent years. The duration of the Cinder Cones event will be determined in following field seasons.

The causes of SSWS are not well understood and lesions that appear similar to those of SSWS can be caused by mechanical injury, for example from predation [8]. Mechanical injury is an unlikely explanation for the McMurdo Sound events, however; there were no obvious evidence of ice scour, which frequently leaves obvious ruts in the seafloor and removal of benthic communities, in the winters of 2019 or 2022. Likewise, predation, potentially from predatory snails, other starfish, or seals, seems unlikely to damage so many animals simultaneously in a restricted area. Anomalously warm water temperatures are often associated with SSWS [5,8,11,17–19], but a year-long benthic temperature record from the McMurdo Intake Jetty shows no unusual warming in 2019 prior to the first observations of SSWS in October [38]. Similarly, annual temperature records from several sites around McMurdo Sound show no anomalously warm temperatures in the year prior to the Cinder Cones outbreak in 2022 (Moran et al., in prep).

One possible cause is suggested by two unrelated and somewhat unusual events at the Jetty and Cinder Cones prior to the outbreaks at each site. At the Intake Jetty in 2018, dive teams performed a major cleaning of biofouling organisms from the seawater intake pipe for the McMurdo Station water system. This pipe is located on the south side of the Jetty just above the impacted area, and the cleaning released a large amount of organic material into the environment (Rob Robbins and Steve Rupp, pers. comm.). The outbreak at Cinder Cones was spatially localized to the area of a methane seep, and the highest rates of disease were in the most active area of the seep. Both the pipe cleaning at the Intake Jetty and the methane seep at Cinder Cones could have caused a large increase in benthic microbial activity due to the abundance of organic material. Methane seeps are also known to be significant sinks of oxygen [39]. This potentially fits with one of the leading hypotheses for the cause of SSWS lesions: that increased organic input leads to increased microbial activity at the epidermal-seawater interface and subsequent deoxygenation of the animal's diffusion boundary layer [7,21]. However, the Cinder Cones methane seep has been present since 2011 [40] and the intake pipe at the Jetty had been cleaned previously in 2002 without any subsequent observations of SSWS (R. Robbins and S. Rupp, pers. comm.). Long-term monitoring for SSWS symptoms, concurrently with measurements of seawater oxygen concentrations, microbial activity, and seawater temperature at seep sites and areas of high human impact are needed to determine the factors that drive these concerning outbreak events.

Through changes in community dynamics due to the removal of important predators, SSWS can have far-reaching effects on benthic marine ecosystems [5,10,13,14]. The important ecological role of *O. validus* [24,25,35] along with the typically slow growth and development of Antarctic organisms [41–43] including *O. validus* [34], suggests that large outbreaks of SSWS will have strong and long-lasting impacts on the nearshore benthos in the Antarctic. The only other previously reported cases of SSWS occurred in one of the warmest and most temperature-variable regions of the Southern Ocean, and yet, the outbreaks we report here in McMurdo Sound occurred in one of the coldest and least variable regions. Thus, modern seawater temperatures in the high Antarctic do not protect against SSWS, and continued warming of the Southern Ocean is likely to increase the severity and prevalence of outbreaks in part by increasing benthic oxygen demand.

## Supporting information

**S1 Table. Survey metadata.** All count, percentage, and density data for from surveys of *Odontaster validus* at the McMurdo Intake Jetty and Cinder Cones.
(XLSX)

## Acknowledgments

Antarctic Service Contract diver R.G. Robbins was the first to see SSWS at the McMurdo Intake Jetty in 2019 and supplied data on diving history at McMurdo Station. We thank both R.G. Robbins and S. Rupp for essential help with surveys at the McMurdo Intake Jetty in 2019 and 2020, as well as observations at other sites. R. Robbins performed all surveys at Cinder Cones in 2022. R. Robbins and S. Rupp made all observations at Granite Harbor, New Harbor, and Delbridge Seamount in 2022. We also thank McMurdo Station support staff for facilitating this research along with team members M.W. Aaron Toh, Graham Lobert, and L. Ardor Bellucci for assisting with the research.

## Author Contributions

**Conceptualization:** Amy L. Moran, Andrew R. Thurber.

**Data curation:** Amy L. Moran.

**Formal analysis:** Amy L. Moran.

**Funding acquisition:** Amy L. Moran, Andrew R. Thurber.

**Investigation:** Amy L. Moran, Rowan H. McLachlan, Andrew R. Thurber.

**Methodology:** Amy L. Moran, Andrew R. Thurber.

**Resources:** Amy L. Moran, Andrew R. Thurber.

**Visualization:** Amy L. Moran, Rowan H. McLachlan, Andrew R. Thurber.

**Writing – original draft:** Amy L. Moran.

**Writing – review & editing:** Amy L. Moran, Rowan H. McLachlan, Andrew R. Thurber.

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
