## [Decision Letter · Decision Letter 0]

30 Mar 2023

PONE-D-23-04532Sea Star Wasting Syndrome Reaches the High Antarctic: Two Recent Outbreaks in McMurdo SoundPLOS ONE

Dear Dr. Moran,

Thank you for submitting your manuscript to PLOS ONE. After careful consideration, I came to the conclusion that the submitted manuscript needs some minor revision.

Please revise your manuscript carefully according to the reviewer's suggestions and provide detailed replies in the rebuttal letter.

We look forward to receiving your revised manuscript.

Kind regards,

Tobias B. Grun, Ph.D.

Academic Editor

PLOS ONE

Journal Requirements:

"Antarctic Service Contract diver R.G. Robbins was the first to see SSWS at the McMurdo Intake Jetty in 2019 and supplied data on diving history at McMurdo Station.  We thank both R.G. Robbins and S. Rupp for essential help with surveys at the McMurdo Intake Jetty in 2019 and 2020, as well as observations at other sites. R. Robbins performed all surveys at Cinder Cones in 2022. R. Robbins and S. Rupp made all observations at Granite Harbor, New Harbor, and Delbridge Seamount in 2022.We also thank McMurdo Station support staff for facilitating this research along with team members M.W. Aaron Toh, Graham Lobert, and L. Ardor Bellucci for assisting with the research. This work was funded by NSF-OPP-1745130 to ALM and NSF-OPP-2046800 to ART.":

"Funded by NSF-OPP-1745130 to ALM and NSF-OPP-2046800 to ART. United States National Science Foundation Office of Polar Programs, https://www.nsf.gov/div/index.jsp?div=OPP. The funders had no role in study design, data collection and analysis, decision to publish, or preparation of the manuscript."

5. We note that Figures 2 and 3 in your submission contain copyrighted images. All PLOS content is published under the Creative Commons Attribution License (CC BY 4.0), which means that the manuscript, images, and Supporting Information files will be freely available online, and any third party is permitted to access, download, copy, distribute, and use these materials in any way, even commercially, with proper attribution. For more information, see our copyright guidelines: http://journals.plos.org/plosone/s/licenses-and-copyright.

a. You may seek permission from the original copyright holder of Figures 2 and 3 to publish the content specifically under the CC BY 4.0 license. 

Reviewers' comments:

Reviewer's Responses to Questions

**Comments to the Author**

1. Is the manuscript technically sound, and do the data support the conclusions?

Reviewer #1: Yes

2. Has the statistical analysis been performed appropriately and rigorously? 

Reviewer #1: No

3. Have the authors made all data underlying the findings in their manuscript fully available?

Reviewer #1: Yes

4. Is the manuscript presented in an intelligible fashion and written in standard English?

Reviewer #1: Yes

5. Review Comments to the Author

Reviewer #1: This manuscript provides important information on the presence of SSE in a key Antarctic asteroid and support its publication. Overall, the manuscript is a straightforward read, although is a bit too descriptive. I have suggestions that the authors should consider.

1. There should be some statistical analysis comparing data between years.

2. L. 145, 179 mentions density. Densities are not provided and these data should be. Are there previous studies of the density of Odontaster in the region to compare with?

3. For all of the %’s provide the sample size that this is based on.

4. What other sea stars are present? It will be good to name them and indicate that they were not infected.

5. There is a new issue of Biological Bulletin devoted to SSW – I suggest some of the papers in this issue should be consulted and cited as the most up to date.

6. PLOS authors have the option to publish the peer review history of their article (what does this mean?). If published, this will include your full peer review and any attached files.

Reviewer #1: No

---

## [Author Response · Author response to Decision Letter 0]

27 May 2023

Editorial and reviewer comments and responses

We have followed these style requirements to the best of our ability. 

We have added the following text to the MS (lines 86-89 in the marked-up manuscript): “No organisms were taken during this project. No permits were required for site access because the study sites were not within any Antarctic Specially Protected Areas (ASPA) or Antarctic Specially Managed Areas (ASMA) as defined by Annex V to the Environment Protocol, Agreed Measures for the Conservation of Antarctic Fauna and Flora, Antarctic Treaty Consultative Meeting.” 

"Antarctic Service Contract diver R.G. Robbins was the first to see SSWS at the McMurdo Intake Jetty in 2019 and supplied data on diving history at McMurdo Station. We thank both R.G. Robbins and S. Rupp for essential help with surveys at the McMurdo Intake Jetty in 2019 and 2020, as well as observations at other sites. R. Robbins performed all surveys at Cinder Cones in 2022. R. Robbins and S. Rupp made all observations at Granite Harbor, New Harbor, and Delbridge Seamount in 2022.We also thank McMurdo Station support staff for facilitating this research along with team members M.W. Aaron Toh, Graham Lobert, and L. Ardor Bellucci for assisting with the research. This work was funded by NSF-OPP-1745130 to ALM and NSF-OPP-2046800 to ART.":

"Funded by NSF-OPP-1745130 to ALM and NSF-OPP-2046800 to ART. United States National Science Foundation Office of Polar Programs, https://www.nsf.gov/div/index.jsp?div=OPP. The funders had no role in study design, data collection and analysis, decision to publish, or preparation of the manuscript."

We have removed the funding-related text from the manuscript (strikethrough in lines 252-253, marked-up copy). No changes are needed to the Funding Statement.

We recommend that you contact the original copyright holder with the Content Permission Form (http://journals.plos.org/plosone/s/file?id=7c09/content-permission-form.pdf) and the following text: “I request permission for the open-access journal PLOS ONE to publish XXX under the Creative Commons Attribution License (CCAL) CC BY 4.0 (http://creativecommons.org/licenses/by/4.0/). Please be aware that this license allows unrestricted use and distribution, even commercially, by third parties. Please reply and provide explicit written permission to publish XXX under a CC BY license and complete the attached form.”

Figure 1 is compliant with the CC BY 4.0 license. The map image was obtained from Landsat, Image Mosaic of Antarctica, and is in the public domain: “Terms of Use: These images are in the public domain and can be used freely and without acknowledgement. However, credit to the Landsat Image Mosaic of Antarctica (LIMA) Project is greatly appreciated.”

https://developers.google.com/earth-engine/datasets/catalog/USGS_LIMA_SR#terms-of-use

We have amended the Figure 1 caption to read “The map used in Figure 1 is modified from the Landsat Image Mosaic of Antarctica (LIMA) Project and is in the public domain.” (lines 93-95, marked-up copy)

5. We note that Figures 2 and 3 in your submission contain copyrighted images. All PLOS content is published under the Creative Commons Attribution License (CC BY 4.0), which means that the manuscript, images, and Supporting Information files will be freely available online, and any third party is permitted to access, download, copy, distribute, and use these materials in any way, even commercially, with proper attribution. For more information, see our copyright guidelines: http://journals.plos.org/plosone/s/licenses-and-copyright.

a. You may seek permission from the original copyright holder of Figures 2 and 3 to publish the content specifically under the CC BY 4.0 license. 

We recommend that you contact the original copyright holder with the Content Permission Form (http://journals.plos.org/plosone/s/file?id=7c09/content-permission-form.pdf) and the following text: “I request permission for the open-access journal PLOS ONE to publish XXX under the Creative Commons Attribution License (CCAL) CC BY 4.0 (http://creativecommons.org/licenses/by/4.0/). Please be aware that this license allows unrestricted use and distribution, even commercially, by third parties. Please reply and provide explicit written permission to publish XXX under a CC BY license and complete the attached form.”

For Figures 2 and 3, we have obtained and uploaded the signed CC BY 4.0 license from the copyright holder. We have amended both figure captions to read “Photographs by Rob Robbins, shared with permission under CC BY 4.0.” (lines 120 and 123-124, marked-up copy)

The caption for the Supporting Information (S1 Table) is at the end of the manuscript, after the references (lines 370-372, marked-up copy). We have amended it slightly to match the new text in the Results: 

S1 Table. Survey Metadata. All counts, percentage, and density data of from surveys of Odontaster validus from surveys at the McMurdo Intake Jetty and Cinder Cones. 

Reviewers' comments:

Reviewer #1: This manuscript provides important information on the presence of SSE in a key Antarctic asteroid and support its publication. Overall, the manuscript is a straightforward read, although is a bit too descriptive. 

We appreciate these comments!

I have suggestions that the authors should consider.

1. There should be some statistical analysis comparing data between years.

We have added a chi-square test showing that the incidence of disease symptoms at the Intake Jetty was significantly lower in 2020 than in 2019.

The following text was added to the Methods (lines 113-118, marked-up copy):

“A chi-square test of independence was performed to compare the frequencies of asymptomatic animals (stage 0) to symptomatic animals (pooled stages 1-4) at the McMurdo Intake Jetty between 2019 and 2020. The same transect lines were surveyed twice in 2019, so to avoid potential double-counting of animals, for 2019 we only used data from the first sampling date on each side of the Jetty (10/17/19 for JS, 10/19/19 for JN). For both 2019 and 2020, data from JN and JS were pooled within year. All data are shown in S1 Table. The analysis was performed in JMP Pro 15.2.0 (SAS Institute Inc., 2019).”

The following text was added to the Results (lines 160-162, marked-up copy):

“A chi-square test showed that there was a significant relationship between disease frequency and year, χ2 (1, N = 909) = 64.24, p <0.0001. Symptomatic animals were significantly less frequent in 2020 compared to 2019.” 

2. L. 145, 179 mentions density. Densities are not provided and these data should be. 

All density data are in S1 Table. To clarify that the data are provided, we have amended the first line of the Results to read “All raw survey count data, percentages, and densities are provided in the supplementary information (S1 Table).” (line 148)

Are there previous studies of the density of Odontaster in the region to compare with?

This is an interesting suggestion, but the density of Odontaster is variable on a small spatial scale at these dive sites, and no previous data are available from our specific transect locations.

3. For all of the %’s provide the sample size that this is based on.

All sample sizes are already given in S1 Table. We feel that presenting count data in a table is much clearer and more readable then presenting the same data in the text, so we have chosen not to duplicate the table data in the text.

4. What other sea stars are present? It will be good to name them and indicate that they were not infected.

We agree this would be interesting to know this, but we do not have these data. We have changed the text to read “Similar to the event at Deception Island, symptoms of SSWS in McMurdo Sound were localized to particular dive sites and were not noted in any other species of asteroid besides O. validus. O. validus was by far the most common species at our sites, though, and other species were not surveyed in a systematic way. Therefore, SSWS in other species cannot be ruled out.” (lines 189-192, marked-up copy) 

5. There is a new issue of Biological Bulletin devoted to SSW – I suggest some of the papers in this issue should be consulted and cited as the most up to date.

We could not find a recent issue of the The Biological Bulletin devoted to SSWS. We did find three papers in the December 2022 issue, which may be what the reviewer meant, but in our opinion these papers did not contain new data or observations that seemed relevant to this manuscript. Without more specific guidance from the reviewer, therefore, we are not sure how to change the MS in response to this comment.

---

## [Editor Report · Decision Letter 1]

4 Jul 2023

Sea Star Wasting Syndrome Reaches the High Antarctic: Two Recent Outbreaks in McMurdo Sound

PONE-D-23-04532R1

Dear Dr. Moran,

We’re pleased to inform you that your manuscript has been judged scientifically suitable for publication and will be formally accepted for publication once it meets all outstanding technical requirements.

Kind regards,

Tobias B. Grun, Ph.D.

Academic Editor

PLOS ONE
---

## [Editor Report · Acceptance letter]

18 Jul 2023

PONE-D-23-04532R1 

Sea Star Wasting Syndrome Reaches the High Antarctic: Two Recent Outbreaks in McMurdo Sound 

Dear Dr. Moran:

I'm pleased to inform you that your manuscript has been deemed suitable for publication in PLOS ONE. Congratulations! Your manuscript is now with our production department. 

Kind regards, 

on behalf of

Dr. Tobias B. Grun 

Academic Editor

PLOS ONE